# ReSkin: versatile, replaceable, lasting tactile skins

**Raunaq Bhirangi**[*], **Tess Hellebrekers**[*], **Carmel Majidi, Abhinav Gupta**

https://reskin.dev/

**Abstract:** Soft sensors have continued growing interest because they enable both passive conformal contact and provide active contact data from the sensor properties. However, the same properties of conformal contact result in faster deterioration of soft sensors and larger variations in their response characteristics over time and across samples, inhibiting their ability to be long-lasting and replaceable. ReSkin is a tactile soft sensor that leverages machine learning and magnetic sensing to offer a low-cost, diverse and compact solution for long-term use. Magnetic sensing separates the electronic circuitry from the passive interface, making it easier to replace interfaces as they wear out while allowing for a wide variety of form factors. Machine learning allows us to learn sensor response models that are robust to variations across fabrication and time, and our self-supervised learning algorithm enables finer performance enhancement with small, inexpensive data collection procedures. We believe that ReSkin opens the door to more versatile, scalable and inexpensive tactile sensation modules than existing alternatives.

## 1 Introduction

In recent years, AI has advanced significantly from large-scale recognition to defeating human players in games. But surprisingly current approaches still struggle at one task: dexterous manipulation. While babies, from a young age, can perform several challenging manipulation tasks, robots continue to struggle even with simple tasks. Why is that? We believe a significant bottleneck in dexterous manipulation is the lack of practical solutions to tactile sensing. From collecting large-scale rich contact data in the wild for learning models to building individual tactile sensors for robot fingers and hand surfaces, current tactile sensing solutions lack on multiple dimensions and fail to scale up.

In the context of robotics and AI, good tactile skins aim to provide: (a) conformal contact for stable grasping/manipulation; (b) accurate compression and shear force measurements; (c) high force (<0.1 N) and temporal resolution (>100 Hz); and (d) large surface area coverage (>4 cm$^2$) with good spatial resolution for sensing at all contact points. For practical usage, good tactile sensors should also prioritize being (e) compact and versatile, (f) inexpensive, and (g) long-lasting. Current solutions for tactile sensing have not been able to address all of these needs. For example, vision-based tactile sensors are often bulky, expensive, and slow to respond (30-60 Hz) [1, 2]. Resistive and capacitive soft sensors require many connections that lead to early failure and integration challenges [3, 4]. Commercial sensing options, such as BioTac, are expensive (>$1000) and available in limited form factors. Rigid tactile sensors, such as force-sensitive resistors, lack the soft, deformable surface that is advantageous for object/environment interaction. Above all, while there has been a plethora of work focused on fingertip sensing, all-over sensing skins are much less studied.

There are two primary reasons why sensing skins have not been practical solutions for tactile sensing: (a) first, there is a direct trade-off between the soft materials that enable conformal contact and their ability to perform well over time. The exact properties that make soft sensors ideal for dexterous manipulation, make them degrade easily during robotic tasks; (b) but more importantly, even skins with durable lasting materials require data-driven modeling which generally fails to generalize from one sensor to another. Therefore, any replacement of skin requires relearning the model which is impractical (hence, limiting experiments to one sensor only [5]).

---

[*]equal contribution

5th Conference on Robot Learning (CoRL 2021), London, UK.

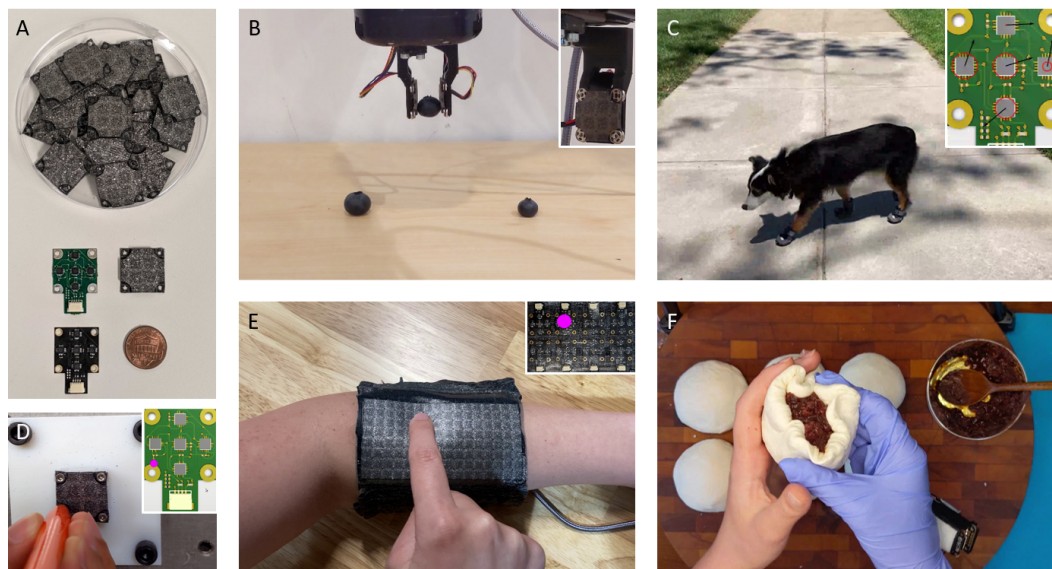

Figure 1: A) ReSkin is easy to fabricate and the size of a penny, enabling a wide range of applications. B) Robot gripper using tactile feedback from ReSkin sensors to hold a blueberry without squishing it. C) Dog shoe with an embedded ReSkin sensor; (inset) visualization of sensor measurements. D) Contact localization on a new ReSkin sensor using our self-supervised adaptation procedure. E) Contact localization on a ReSkin curated into a fabric sleeve as a 2in x 4in contiguous skin. F) ReSkin sensor as a fingertip sensor to record forces and contacts while folding a dumpling

We propose ReSkin – an inexpensive (<\$30), replaceable, compact, versatile and long-lasting tactile soft skin. ReSkin is composed of soft magnetized skin and a flexible magnetometer-based sensing mechanism. Any deformation of the skin caused by normal/shear forces is read via distortions in magnetic fields. These distortions can be mapped back to estimate the contact points and forces on the original skin using a learned machine learning model. The ReSkin design is compact (2-3mm thick) and long-lasting (our ML models perform accurate predictions even beyond 50K interactions. ReSkin is versatile – the skin and the sensor mechanism can be used anywhere from robot hands to objects to gloves, arm sleeves and even dog paws. ReSkin has high temporal (up to

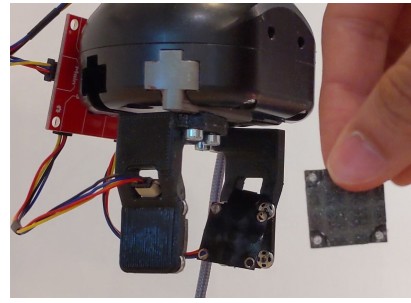

Figure 2: ReSkin is replaceable!

400Hz) and spatial resolution (1mm with 90% accuracy). But what makes ReSkin the ideal tactile skin is the ability to replace an old skin with a new skin as if you are peeling off an old band-aid and putting a new one on. Our learned models perform strongly even on new skins out-of-the-box but can be further adapted to high precision and resolution using a self-supervised calibration technique. We believe ReSkin has the ability to collect contact data in the wild, provide robust tactile perception capabilities to our robot (See Figure 1) and effectively make tactile perception a first-class citizen among its peers (pixels and sound).

## 2   Background

Soft sensing skins provide tactile or proprioceptive information without affecting the underlying mechanics of the system. Machine learning approaches have recently been shown to effectively parse soft sensor data for robotic grasping [6, 7, 8], proprioception [9], and object classification [10], among many others [11, 12, 13]. Recent work in this area has relied on all types of sensing principles to collect and infer information in all types of form factors [14, 15, 16]. For example, resistive networks patterned onto a knit glove have been used to collect tactile grasping dataset and classify objects using neural networks [17]. Capacitive soft skins can be scaled up to larger areas by sampling at different interrogation frequencies to detect position via SVMs with just one interface [18]. Multi-modal sensing skins have also been shown to improve the ability to discern between 8 types

of applied deformation using neural networks [19] and static or dynamic inputs [20]. Data-driven approaches are becoming more common over traditional modeling due to the unpredictable material properties that introduce both non-stationary responses and non-linear behaviors over time, especially considering the dynamic interactions and unconstrained environments robots may encounter.

Unlike capacitive[18], resistive[17], and piezoelectric[21] soft sensing, magnetic and optical soft sensors do not require direct electrical connections between the circuitry and elastomer. This is ideal to keep cost down, as the elastomeric interface degrades much faster than the accessory electronics. It also simplifies the replacement process by not requiring the user to disconnect and reconnect individual wires. While optical sensors provide high spatial resolution data, they also require a clear line of sight between the camera and elastomer to observe deformations[22, 23]. The camera's depth-of-focus puts a hard limit on the minimum distance to the elastomer surface leading to relatively bulky sensor modules. In contrast, magnetic sensing benefits greatly from minimizing the distance between sensor and elastomer, allowing for a much more compact tactile sensor. In addition, the small form factor of magnetometers, as compared to cameras, enables compatibility across more diverse form factors for the tactile sensor. We demonstrate these key benefits by integrating the magnetic skin onto an arm sleeve, glove, dog shoe and a robot end-effector. In each case, the elastomer is removable while the circuitry stays in place. While there have been a number of attempts towards developing a large area skin - piezoresistive fabrics [16, 21, 24], rigid taxels [25] as well as optical sensors[26], they often lack shear sensing capabilities[25, 21, 24], conformal contact[25] and/or a scalable fabrication process[21]. ReSkin, however, is uniquely positioned to satisfy all of these requirements, and has the potential to be a scalable solution for all-over sensing skin.

| | ReSkin | DIGIT[1] | GelSlim [22] | BioTac [27] | RSkin [21] |
|---|---|---|---|---|---|
| Type | Magnetic | Optical | Optical | MEMS | Piezoresistive |
| Frequency | **400Hz** | 60 Hz | 60 Hz | 100 Hz | ? |
| Variable Form Factor | ✓ | ✗ | ✗ | ✗ | ✓ |
| Thickness <3mm | ✓ | ✗ | ✗ | ✗ | ✓ |
| Low Cost | ✓ | ✓ | ✓ | ✗ | ✗ |
| Easily replaceable | ✓ | ✓ | ✓ | ? | ✗ |
| Area coverage | ✓ | ✗ | ✗ | ✗ | ✓ |
| Durable (>50k contacts) | ✓ | ? | ✗ | ✓ | ? |

Table 1: ReSkin is the only sensor that satisfies all the requirements for learning approaches

The underlying principle for magnetic sensing is that an applied deformation is measured as a change in magnetic flux readings by nearby magnetometers. However, we still need to learn or estimate the mapping function that decodes change in magnetic flux into contact force position and magnitude. Several works on soft sensors have used neural networks for sensor characterizations [28, 29], but these models are often trained on single sensor prototypes, and do not necessarily transfer to new copies of the sensor. Then, the end-user is required to collect and sometimes label their own data for each sensor, which additionally requires access expensive, specialized equipment [29, 30].

In this paper, we systematically perform an experimental analysis of the proposed magnetic tactile sensor. First, we extensively study the characterization of a single sensor over time. We demonstrate that one can learn a quite accurate data-driven model to map magnetic flux changes to contact force location and magnitude. We also demonstrate that the skins are long-lasting. However, models trained on one sensor fail to generalize to other sensors or to different circuit board designs. Our first insight is to exploit multi-sensor learning: learn a more generalizable model by using data from a larger number of sensors. While this leads to significant improvement, it still falls well short of training and testing on the same sensor. Inspired by recent work in self-supervised learning, we also present a simple self-supervised calibration procedure which learns to adapt the multi-sensor model to a particular sensor using just a couple of hundred pokes on the skin. Our self-supervised approach is inspired from several works in slow feature learning [31, 32] and contrastive learning [33, 34, 35].

## 3 Design and Fabrication

The sensing principle for ReSkin relies on relative distance changes between embedded magnetic microparticles in an elastomer matrix and a nearby magnetometer. The use of magnetic microparticles allows the skin to be molded into many shapes and thicknesses. When the magnetic composite is deformed by applied force, the magnetometer reports changes in magnetic flux in its X-,Y-, and

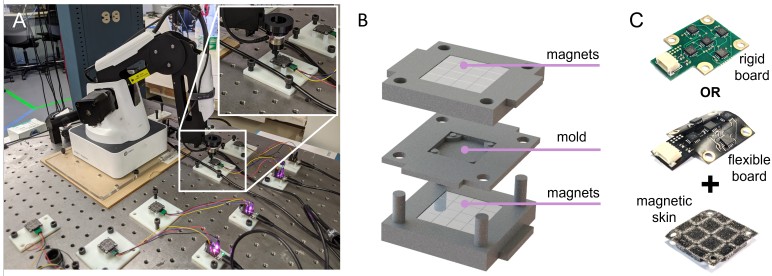

Figure 3: A) Experimental setup for data collection with Dobot Magician, ATI Nano 17 (inset), and six sensor boards streaming to a control computer. B) Mold for curing elastomer along with magnet holders. C) Two types of circuit boards – rigid and flexible – designed to be used with ReSkin.

Z- coordinate system [36]. For an overall sensing area of 20mm x 20mm (Figure 3), we measure magnetic flux changes using 5 magnetometers. Four magnetometers (MLX90393; Melexis) are spaced 7mm apart around a central magnetometer. All 3D-printed molds, circuit board files, bill of materials, and libraries used have been publicly released and opensourced on the website.

**Magnetic Elastomer Fabrication.** Our fabrication technique takes advantage of strong edge-effects of permanent magnets. We magnetize the composite over a 4x4 grid of much smaller cube magnets (0.25in; AppliedMagnets). Additionally, we apply a more uniform magnetic field during curing by placing the grid of magnets both above and below the sample (Figure 3B). Magnetic microparticles (MQP-15-7; Magnequench) and 2-part polymer (Dragonskin 10 NV; Smooth-On) are mixed in a 2:1:1 ratio and poured into a 3D-printed mold. The mold is placed in a vacuum chamber for 3 minutes to remove air bubbles before placing magnets above and below the sample. The sample cures at room temperature (approx 24°C) for at least 3 hours before being removed.

**Circuit Board.** We use two variants of the circuit board for the experiments and demos presented in this paper – a rigid board and a flexible board illustrated in Figure 3C. The circuit includes a 4-pin connector (JST-SH; Molex) that transfers 20 values of magnetometer data (Temp, $B_X, B_Y, B_Z$ for 5 chips) to a microcontroller (Trinket M0; Adafruit) at approximately 400 Hz. The microcontroller processes and transmits this data over USB to be read over serial from a central computer. To allow for easy replacement without damaging the board, we avoid the use of permanent adhesive for attaching the sensing skin. Skins are secured using four screws (M2-6; for the rigid boards), or using four sewable snaps (size 4/0, Dritz; for the flexible boards). The bottoms of the rigid boards are first insulated by applying a thin layer of polymer (nitrocellulose).

## 4 Experimental Setup

Our goal is to perform experimental analysis of the proposed sensor, ReSkin, and the learned mapping between the magnetic field measurements from the sensor ($\mathbf{B}$) and the planar location ($\mathbf{x} = (x, y)$) and magnitude of the applied force ($\mathbf{F}$). We want to analyze how ReSkin performs on the different desired attributes – the accuracy of contact force estimation, spatial resolution of contact prediction, robustness to wear and tear and how model performance varies across different skin instances. For all the experiments, we use the data collection apparatus shown in Figure 3. The circuit board along with the skin is fixed to a 3D printed mount and streams 4 values, (Temp, $B_X, B_Y, B_Z$), measurements for each of the five magnetometers at 400 Hz. A hemispherical indenter fastened to the end of a Dobot Magician robot is used to apply forces at different locations on the skin. The indenter also encases a 3-axis F/T sensor (Nano17; ATI) that streams force data at 1kHz.

We restrict ourselves to quasi-static measurements and analysis for the results presented in the following sections, unless stated otherwise. The world coordinate frame used in these experiments is defined such that the xy-plane is aligned with the base of the robot as shown in Figure 3. To collect data, we first specify a location for the robot to move to such that the indenter makes contact with the skin. We then record five measurements from the sensor board. The specified xy-location is used as the ground truth label for the location of the force. The normal force measured by the Nano17 sensor is used as the ground truth label for the magnitude of the applied force.

The indentations are made in a snake-like pattern along a 9x9 grid (excluding 4 points per corner; a total of 65 indentations) of size 16cm x 16cm shown in Figure 3. During each iteration, we do a

single pass at each of 6 depths from 0.2mm to 1.2mm, for a total of 390 indentations. We collect data over multiple iterations.

## 5  Single Sensor Model – Decoding Magnetic Flux to Contact Characteristics

Our first experiment is to evaluate the accuracy of the mapping from magnetic flux $\mathbf{B}$ to contact force location and magnitude prediction. Our five-layer multilayer perceptron (MLP) architecture for the mapping function is: `B(15)` $\rightarrow$ `MLP+ReLU(200)` $\rightarrow$ `MLP(200)` $\rightarrow$ `MLP(40)` $\rightarrow$ `MLP+ReLU(200)` $\rightarrow$ `MLP+ReLU(200)` $\rightarrow$ `xyF(3)`. The change in magnetic field resulting from deformation is used as the input to our model. The third activation layer is the bottleneck feature layer. We use $feat(\cdot)$ to represent this 3-layer feature extraction network. Our loss function is L2-loss on $(x, y, F)$. We define the accuracy of contact localization as the fraction of points whose $x$ and $y$ predictions are both within $\pm 1$ mm of their true label. We collect a total of 50K samples and use a random 45K for training and 5K for test. On this simple experiment, we get MSE for location and force is $0.037 \pm 0.014$ mm$^2$ and $0.005 \pm 0.002$ N$^2$ respectively, with a contact localization accuracy of $99.58 \pm 0.34\%$.

To demonstrate the shear sensing capability of the sensor, we perform another experiment. Instead of the quasi-static setup explained in Sec. 4, data is collected dynamically by indenting the skin to a certain depth and dragging it along the length of the sensor. We move in straight lines along x and y directions at intervals of 2 mm to cover the entire area of the sensor. The network architecture is the same as described in the previous paragraph, predicting $(x, y, F_x, F_y, F_z)$ instead of just $(x, y, F_z)$. On this experiment, we get MSE for $F_{xy}$: $0.0011 \pm 0.0002$ N$^2$, without compromising on prediction of normal force (MSE:$0.003 \pm 0.001$ N$^2$) or contact location (MSE:$0.085 \pm 0.006$ mm$^2$).

Of course, the above setup is not realistic since training and test data is unlikely be sampled randomly from the same distribution. Instead, we use a more practical setting, training on an initial $K$ samples and testing on the samples that come after. As the elastomer goes through multiple cycles of compression and retraction, we see a drift in the properties of the elastomer. This is evidenced by the variation in the recorded magnetic field shown in Figure 4a. Therefore, it is critical to analyze how the learned model behaves with respect to time.

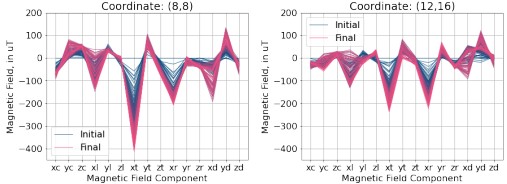 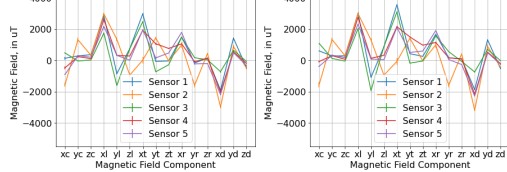

(a) Variation of magnetic field over time for a single sensor at two different points, over 10,000 indentations. The first measurement is subtracted from other measurements to better illustrate degree of variation.

(b) Variation in magnetic field at two different points across five different sensors. Each line corresponds to the average magnetic field measured over 10,000 indentations for a particular sensor

Figure 4: Variation in magnetic field over time and across different sensors. Each tick on the x-axis corresponds to a component of the magnetic fields measured by the sensor. While the general properties of the individual sensors overlap, there is still obvious variation across the samples.

Since we learn a sensor model that uses the change in magnetic field as input, we would need to record the magnetic field before and after contact occurs. Depending on the application scenario in which the sensor is deployed, it may often be easier to collect calibrating no-load magnetic field measurements at regular intervals. Here, we design an experiment to quantify the effect of the frequency of this measurement on learned sensor models. We collect data from 50,000 indentations on a single sensor. We train a neural network to predict contact location and force, using the first 5,000 indentations as the training set. This model is then evaluated on test sets comprising 1000 indentations after every 5000 indentations, to understand the degree and rate of domain shift. The results of this experiment are shown in Figure 5.

Based on Figure 5a, we observe that prediction errors are higher and increase faster when we only make a single no-load measurement at the start. This can be attributed to the drift in the elastomer response over time, which can be offset to a certain extent by more frequent no-load measurements. Errors are also higher when no-load measurements are updated before every contact. This could be

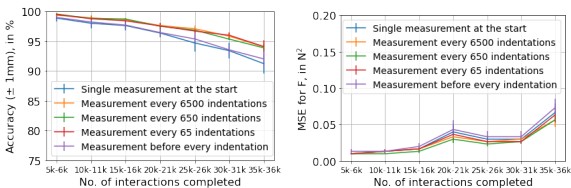
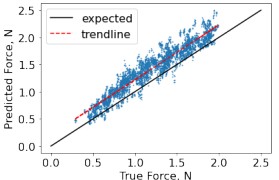

(a) Accuracy of contact localization and MSE for force predictions, as the number of interactions with the skin increases

(b) Scatter plot of predicted force and the actual force applied, after 45,000 interactions.

Figure 5: Model performance with increasing number of interactions

a result of overfitting to the training data, since later experiments were seen to significantly benefit from updating zero measurements just before every contact. Furthermore, Figure 5b indicates that the model, on average, overestimates the applied force. This overestimation can be attributed to the softening of the elastomer as the number of interactions increases.

## 6 Adapting to New Sensors – MultiSensor Model + Self-supervised Learning

Our goal is to provide a simple, replaceable tactile sensor. To achieve this, it is imperative that any learned models acting on sensor measurements generalize to new sensor boards and skins. We demonstrate the generalizability of our learned sensor response model in the following sections. Our models predict the contact normal force ($F$) and location($\mathbf{x} = (x, y)$) using the change in magnetic field measured by the magnetometers($\mathbf{B}$). However, the cheap and easy fabrication method for ReSkin comes with a significant degree of variability in the sensor response. Figure 4b demonstrates the variation in raw magnetic field resulting from an identical indentation across different sensors. So, how can we learn a model that generalizes to new sensors and even new circuit boards?

We use two techniques to help improve generalization for new skins and PCBs. First, instead of using data from a single sensor, we use data from multiple sensors to train our mapping function. This allows the model to see more diverse data in training and learn a more generalizable mapping function. Additionally, we apply a feature regularization component (self-supervised loss) to our loss function. This component is a triplet loss computed in feature space as follows:

$$\mathcal{L}_{\text{triplet}} = \max\left(0, ||feat(\mathbf{B}_a) - feat(\mathbf{B}_p)||^2 - ||feat(\mathbf{B}_a) - feat(\mathbf{B}_n)||^2\right), \tag{1}$$

where $\mathbf{B}_a, \mathbf{B}_p$ and $\mathbf{B}_n$ are three datapoints with corresponding contact locations $\mathbf{x}_a, \mathbf{x}_p$ and $\mathbf{x}_n$, such that $||\mathbf{x}_a - \mathbf{x}_p|| < ||\mathbf{x}_a - \mathbf{x}_n||$, ie. $\mathbf{x}_a$ is closer to $\mathbf{x}_p$ than $\mathbf{x}_n$. Subscripts $a$, $p$ and $n$ refer to anchor, positive and negative samples respectively. This loss encourages points that are closer on the skin to be closer to each other in feature space. It acts as a regularizer while also enabling us to use the self-supervised adaptation procedure described in the following paragraph.

Note that this self-supervised loss does not require ground-truth contact location or force readings and therefore can be leveraged to further improve performance on new sensor boards and skins. A new user can collect their own unlabeled dataset, which can be indexed without requiring explicit labels. For instance, the user can use the tip of a pen to indent the sensor skin in a straight line and incrementally index these points as they move along the line. Triplets of points can now be sampled along this line, and the indices can be used to order the pairs within each triplet by distance. Our multi-sensor learned model can then be fine-tuned using these triplets to minimize the triplet loss. At every training step, we sample a batch from the original training data, and an equal-sized batch of triplets (sampled with replacement) from the unlabeled dataset. The former is used to minimize the original loss function, while the latter is only used to minimize the triplet loss.

### 6.1 Results

We compare four model approaches. Our baseline model is trained on one sensor and tested on a different sensor. The other three are multi-sensor models – (a) trained without the triplet loss, (b) trained with the triplet loss, and (c) trained with the triplet loss along with self-supervised adaptation.

For the following comparisons, we collect data of 10,000 indentations each from 18 different skins. We use a set of 6 sensor boards and 18 skins: each board appearing thrice in the dataset, each time with a different skin on top. For the multi-sensor models, we perform 6-fold cross-validation, with the held out test set corresponding to three different skins on a particular sensor board each time. For the self-supervised adaptation, we use different subsets of the unseen sensor data in the

| Model | Accuracy, in % | $MSE_{xy}$, in mm$^2$ | $MSE_F$, in N$^2$ |
|---|---|---|---|
| Single-sensor | 25.24±10.12 | 6.453±3.363 | $0.420 \pm 0.149$ |
| Multi-sensor without triplet loss | 84.43±12.88 | 0.733±0.707 | $0.155 \pm 0.025$ |
| Multi-sensor with triplet loss | 81.03±12.86 | 0.756±0.718 | $0.155 \pm 0.030$ |
| Multi-sensor with triplet loss, adapted using 390 indentations | $\mathbf{87.00 \pm 11.81}$ | $\mathbf{0.514 \pm 0.601}$ | $\mathbf{0.142 \pm 0.025}$ |

Table 2: The single-sensor baseline performs poorly, failing to capture variability across sensors. Our self-supervised adaptation significantly improves prediction accuracy as well as MSE in xy, F adaptation step, to qualitatively illustrate the effect of the quantity of data used for adaptation. For the single-sensor model, we individually train on 3 different sensors and test on 9 other sensors.

Based on Table 2, we see that the multi-sensor models do significantly better than the single-sensor model. Training on a larger set of sensors allows the neural network model to generalize better. Further, we see that adding the triplet loss slightly affects performance. This small drop could be attributed to the additional constraint on the feature space resulting from the triplet loss. However, the self-supervised adaptation gives us a sizeable improvement over the model predicting without adaptation. Note that the adaptation procedure also results in improved force prediction performance.

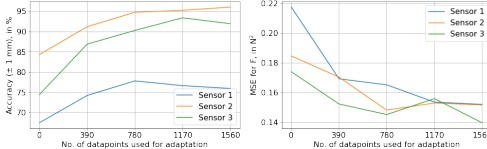 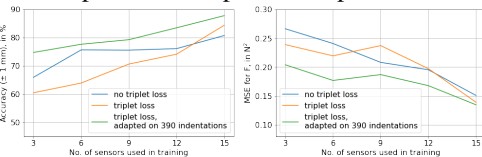

(a) Self-supervised adaptation improves significantly even with small quantities of adaptation data

(b) Self-supervised adaptation leads to even larger performance gain with fewer sensors in training data

Figure 6: Self-supervised adaptation works with lesser adaptation data as well as training data

To further investigate the effectiveness of our self-supervised adaptation procedure, we look at how increasing the quantity of data used for adaptation correlates with model performance. As can be seen from Figure 6a, there is a significant improvement in performance between no self-supervision and using 780 points for self-supervision. However, performance seems to plateau as we increase the data beyond this point, which indicates that our model adapts quickly with small amounts of unlabeled data. Finally, Figure 6b shows the effect of the number of sensors in the training data on model performance. We use data from three unseen sensors as the test set. As expected, performance improves as the amount of training data increases. Further, we observe that the self-supervised adaptation, while always doing better than the other models, offers a significant improvement in performance when using fewer sensors in training.

**Generalizing models with manual indentations** Since the analysis of our self-supervised adaptation technique is performed in a very controlled scenario we perform another, less controlled, experiment which is likely to be closer to the end user's application setting. We manually indent the sensor 325 times and run self-supervised adaptation using the triplet loss. To test the effectiveness of the adaptation, we collect test data using the original experimental setup to evaluate the adapted models. After adaptation we see improvements in accuracy, from 79.86% to 84.84%, $MSE_{xy}$ from 0.676 mm$^2$ to 0.489 mm$^2$ and $MSE_F$ from 0.268 N$^2$ to 0.192 N$^2$, clearly demonstrating the effectiveness of our proposed method even outside a controlled environment.

**Generalizing models to a different sensor board type** In order to demonstrate the effectiveness of our self-supervised adaptation scheme, we now adapt a model learned using rigid sensor boards on a flexible sensor board. Note that this is a significantly harder adaptation problem since the distance between skin and circuit board is 80% lesser in case of the flexible board. We see an average contact localization accuracy of 75% with MSE error on location and force as 0.72 mm$^2$ and 0.54 N$^2$ respectively. The relatively larger force errors can be attributed to an overestimation bias resulting from signals appearing stronger due to the reduced thickness of the flexible board.

## 7 ReSkin in Action

We have demonstrated that ReSkin is a sensor capable of high resolution contact localization and force prediction. The performance does not deteriorate significantly with wear and tear. But

most importantly, learned models generalize to new skins with a simple self-supervised adaptation scheme. We now highlight how ReSkin's compact design allows it to be used in diverse applications with different form factors. In this section, we use ReSkin in different settings to emphasize its effectiveness in a range of application scenarios. These demos are for proof-of-concept only. For the following, we fabricated both flexible and rigid circuit boards of the exact same design: flexible boards are more comfortable and thinner, while rigid boards can withstand larger applied forces.

**Force Sensitivity: Water in shot glass** To visually illustrate the force sensitivity of our sensor, we do a pouring demo where we place a shot glass on top of a ReSkin sensor. As the water fills up, we see monotonically increasing sensor measurements indicating the sensor's ability to distinguish forces as small as the weight of less than 20 mL ($< 0.2$N) of water (Supplementary Video S1).

**Robot Gripper.** Next, we show that ReSkin can be a useful tactile sensor for robotics applications such as grasping delicate objects such as blueberries and grapes (See Figure 1B). Two ReSkin sensors with flexible circuit boards are placed on either side of a parallel jaw gripper (Robotiq Hand-E Gripper on Sawyer Arm). Grasping soft and squishy objects requires force feedback – applying too much force will squish blueberry and the grape. We show that the built-in force sensing (30N minimum) is insufficient for the task, whereas ReSkin does an excellent job of using force feedback to control grasping. Furthermore, we demonstrate that the grasping continues to work well, with no tuning required, when we replace one of the skins with a new skin (Supplementary Videos S2, S3).

**Location Sensitivity: Poking** To visually illustrate the location sensitivity of our sensor, we do a simple poking task on a new sensor and show the resolution of real-time contact location estimation (See Figure 1D and Supplementary Video S4).

**Dog Shoe.** Our next application demonstrates how ReSkin's compact design makes it non-obtrusive and useful for measuring tactile forces in the wild. One magnetic skin and flexible circuit board is placed inside the sole of a dog shoe (size: 1.75in). A 1/16in layer of urethane foam is added on top for comfort. The data is collected on-board and logged to an SD card at 250 Hz. The sensorized shoe is worn on the front right leg of a small dog (17 lb). The sensor tracks magnitude and direction of applied force while resting, walking, and running (See Figure 1C and Supplementary Video S5).

**Glove.** We also demonstrate how ReSkin can be used to measure forces during natural human-object interactions. A magnetic skin and rigid circuit board is placed on the right-hand index finger. A nitrile glove was placed over to hold the board in place, and keep the objects clean. The data was collected on-board and logged to an SD card at 250 Hz. We demonstrate sensor output during the sealing of dough (See Figure 1F and Supplementary Video S6).

**Arm Sleeve.** Finally, we want to demonstrate that ReSkin is a surface sensor and can be used for wide coverage tactile sensing. Specifically, we connected 8 flexible boards in two rows of four and fabricated a larger, continuous skin (2in x 4in). All 8 boards are connected to a microcontroller (QT Py; Adafruit) that samples all 40 magnetometers at 133 Hz. We show how ReSkin can be scaled up for contact localization across larger surface areas (See Figure 1E Supplementary Video S7).

## 8 Conclusion

We present ReSkin: a low-cost, compact and long-lasting surface tactile sensor with high localization accuracy and force sensitivity. ReSkin combines soft sensing with recent advances in machine learning to develop models that generalize across time and individual skins. More specifically, we use multi-sensor learning combined with self-supervised triplet loss for slow feature changes. We also present an SSL adaptation procedure to further refine the models for new skins. Therefore, ReSkin sensors have easily replaceable skin (as easy as peeling and putting new band-aid) that can be used right away. We demonstrate that the compact form of ReSkin makes it an ideal candidate for diverse applications: from grasping delicate objects to measuring forces exerted by dog feet; from building wide-coverage contiguous skin to measuring contact forces in the wild.

**Limitations and Future Work:** While we have shown promising results on contact localization and force prediction, there is enormous untapped potential for ReSkin at this stage. Experiments in this paper are based on single point contact, and we aim to further investigate multi-point contact. An interesting direction for future work is to analyze the effect of external magnetic fields and metallic objects on ReSkin's sensing ability. ReSkin can stream data up to 400 Hz and we aim to leverage this capability to train better models using dynamic time-series data. We believe that ReSkin (and its desirable properties) will make tactile perception far more accessible for real-world use.

## Acknowledgments

The authors would like to thank Sudeep Dasari for his help with setting up the Sawyer robot and Yunsik Ohm and Zach Patterson for their help with fabrication in the initial stages of the project. The authors would also like to thank everyone at AGI Labs as well as the Soft Machines Lab for their constant help and support. CM was supported in part by NSF-NRI #1830362.

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
