# OpenReview forum: "ReSkin: versatile, replaceable, lasting tactile skins"
_robot-learning.org/CoRL/2021/Conference — CoRL2021 Oral_

### Official Review · Reviewer_2SmQ · 2021-07-23

**Originality:** Very Good
**Technical Quality:** Good
**Clarity Of Presentation:** Good
**Impact:** 4

**Recommendation:**

Weak Accept: I recommend accepting the paper, but will not argue for my recommendation if the majority of other reviewers have a different opinion.

**Summary:**

In this paper, the authors present a new type of low cost contact sensor based on magnetic field distortions. The core idea is to use fabricated elastomer materials with embedded magnetic particles. When the material is distorted under pressure, the relative distances between the magnetic particles will also shift. The change in the magnetic flux can be detected by an array of magnetometers. The authors used supervised training to calibrate the contact sensor, i.e. map magnetic field readings to contact location and force magnitude using a neural network. They also proposed a triplet loss to adapt their learned sensor response model to changes in the elastomer material or sensor boards without the need to use labelled new data.  The authors demonstrated that their contact sensor can be used for various applications such as manipulation and locomotion.

**Issues:**

See my comments in the main limitations.

**Reviewer Expertise:**

Good: General knowledge of the area

**Strengths And Weaknesses:**

The main strength of this paper includes:

(1) A fabrication process to produce cheap, replaceable "skins" that can be used to measure contacts.

(2) A learned sensor response model that maps the magnetic flux to contact forces and locations. By combining readings from multiple sensors and regularizing the training using a self-supervision loss, the model can generalize to new materials and sensor boards without complicated calibration process.

(3) Can be applied to many different types of tasks.

The main limitations of the paper are:

(1) Generalization to simultaneously multiple contact points. In the proposed pipeline, the authors used single point contact data (i.e.  "indentation") to train their model. And the validation and adaptation also depends on point contact responses. This raised a question on whether/why their learned model can generalize to a multi-point contact, or surface contact situations. The deformation of the material is a nonlinear response, so one cannot just assume a simple addition of the magnetic flux happens under a continuous surface load. Needs more discussion and experiments to explain and validate the more general situation. For example, Is a ReSkin unit really measuring the center of pressure or something else?

(2) Continuous adaptation. The authors claimed that when a new "skin" is replaced, the model can still generalize by self-supervised training using the triplet loss + old labelled data. The question is if this process will eventually produce bias overtime? For example, what happens if one swaps the elastomer 10 times? After 10 adaptations is there any "drift" in the model so the performance will degrade? Needs more discussion and experiments on this.



**Summary Of Recommendation:**

Given the broad potential application and simplicity of recalibration using self-supervision, I think the ReSkin technique and their calibration model can be potentially very useful for the robotics community in general.

---

> ### Author Response · Authors · 2021-08-20
> **Response to Reviewer 2SmQ**
>
> We thank the reviewer for positive feedback. We answer two of the review concerns below:
>
> 1. __Multi-point and large surface contact:__ Multi-point contact and/or large surface contact are possible but beyond scope of current paper. The deformation to magnetic flux relationship is non-linear but the magnetometer reads a vector sum of the full environment. The authors do believe ReSkin measures the center of pressure. Previous work [1] tested a larger indentor on small indentor dataset and it was seen to work fairly well
>
> 2. __Continuous adaptation:__ We always start from the base model [since magnetometers do not degrade over time] when replacing the skin and therefore, we do not need to worry about any additional drift.
>
> [1] Hellebrekers, T., Chang, N., Chin, K., Ford, M. J., Kroemer, O., & Majidi, C. (2020). Soft magnetic tactile skin for continuous force and location estimation using neural networks. IEEE Robotics and Automation Letters, 5(3), 3892-3898.

---

### Official Review · Reviewer_51C5 · 2021-07-23

**Originality:** Good
**Technical Quality:** Very Good
**Clarity Of Presentation:** Good
**Impact:** 4

**Recommendation:**

Weak Accept: I recommend accepting the paper, but will not argue for my recommendation if the majority of other reviewers have a different opinion.

**Summary:**

The paper proposes ReSkin, an inexpensive, replaceable, and robust tactile skin based on magnetometer-based sensing mechanism and discusses learning methods for calibrating the sensor raw signals to contact localization and force prediction output across different skins and sensing boards.


**Issues:**

See “Suggestions for Improvement” section above.

**Reviewer Expertise:**

Excellent: Expert knowledge on the topic of the paper

**Strengths And Weaknesses:**

Strengths:
- The skin is inexpensive, replaceable, and robust to wear and tear. I really like how easy it is to replace the skin layer
- The skin can sense normal forces and potentially shear forces as well (not shown in this paper)
- The evaluation was done with a large variety of skins
- The demonstrations were shown on a variety of applications

Suggestions for improvement:
- Clarifying contributions with respect to the state-of-the-art: There is an extensive body of work on tactile skins (both large-area and small with resistive, capacitive, magnetic, optical, and multimodal sensing etc.) for robots which have not been cited in this paper. Clarifying the contributions of this paper with respect to this extensive body of work (especially large-area skins which was one of the primary motivations of this paper ( https://spj.sciencemag.org/journals/research/2019/3018568/, https://www.sciencedirect.com/science/article/pii/S0921889014001821,  https://ieeexplore.ieee.org/document/6548392, https://ieeexplore.ieee.org/stamp/stamp.jsp?arnumber=8812712, https://www.sciencedirect.com/science/article/abs/pii/S0921889016307837, https://arxiv.org/abs/1803.00628, https://link.springer.com/chapter/10.1007/978-3-319-04123-0_6, https://ieeexplore.ieee.org/stamp/stamp.jsp?arnumber=9247533 , and many more ...) and other magnetic sensing based skins that the authors have already cited such as Hellebrekers et al.,Yan et al.) would make the paper stronger.

- The paper can also become stronger by discussing the limitations of this sensing skin. Is the sensing technology affected by the presence of other magnetic objects in the environment? Are multiple-touch scenarios possible with this skin (all the demonstrations and data collection were single touch)? Is there any cross-talk when multiple touch points are proximally located? While the data acquisition rate is impressive, the latency is not clear. Analyzing these effects or discussing limitations and potential solutions can make the paper more complete.

- While the evaluation was done with a variety of skins and a lot of data, it is not clear what the insights from the generalization results are. The paper can benefit from details such as how do the 18 different sensing skins differ so that it’s clear what is the generalization across? Do the results generalize across different indenter sizes? In touch modality, sensing is inherently coupled to action and analyzing these effects while considering variations across parameters that affect the actions are crucial. While the results with self-supervised adaptation across multiple sensors are encouraging, in a new scenario, the fine-tuning demonstrated in the paper is done using data collected in a very controlled scenario. Is this feasible in unstructured real-world scenarios?

- Has the paper considered using model-based approaches for calibration and using data-driven methods to model the residuals or unmodeled effects? In the real-world, collecting data is hard especially with contact and conditions vary widely. Having a physics-based model as a prior or to provide some structure can potentially benefit a lot during calibration in previously unseen scenarios.

- Can the paper discuss how (or if) the sensing method can be extended to normal 6D force/torque sensing and shear sensing, which could be very useful in real physical interactions? This could be a strong differentiating factor with many existing large-area tactile sensing skins.

- The paper should be carefully reviewed throughout for typos and word omissions. For example: Line 254: … data, and ?


**Summary Of Recommendation:**

While the paper discusses an encouraging tactile skin technology and learning-based calibration method, it is not clear what the contributions are with respect to other magnetic skins or large-area skins in general. Also, the applicability of the skin and scalability of the calibration procedure needs to be analyzed especially in the light of unstructured scenarios. This is on the way to becoming a strong paper but it needs more work at its current state.

------

The rebuttal has addressed most of my concerns. I am changing my score to Weak Accept.

---

> ### Author Response · Authors · 2021-08-20
> **Response to Reviewer 51C5**
>
> We thank the reviewer for their detailed and constructive comments. We are pleased to report that we have done all the experiments and discussions the reviewer has asked for. This includes shear force, SSL in uncontrolled setup, novelty discussion and finally the limitations. We will include all these in the final version of the paper. As reviewer points out that with these changes, it will be a strong paper -- we hope with these changes the reviewer can confirm and update the rating accordingly.
> 1. __Clarifying contributions with respect to state of the art:__
> The authors agree the paper would benefit from more comparison to related works and clearly outlining novelty. Please see the main comment to AC for discussion on novelty and how we believe our sensor is the top viable option for robot sensing.
>     1. Unlike resistive and capacitive sensors, magnetic and optical tactile sensors do not need to sequentially sample each module to create a full tactile reading. Due to the common grid-like or array-like layout of resistive and capacitive sensors, there are many wires that are required to measure the response. This makes them too fragile and difficult to scale up for robotic sensing.
>
>     2. Optical sensors, such as GelSight and other gelsight-based sensors (GelSlim[1], Gelsight Wedge[2], OmniTac[3] etc), use cameras to image the surface, which doesn’t require any wires. However, cameras require a minimum distance to the surface which results in bulky tactile modules as laid out in the table above. Furthermore, this can result in sensors with a fixed form factor that are very difficult to scale up to other shapes and sizes. The size of the sensor can make it necessary to have intrusive tactile solutions [4] which are often undesirable.
>
>     3. Magnetic sensors: ReSkin differs from previous magnetic tactile sensors by using magnetic microparticles over embedded magnets. This allows reskin to be molded and cut into more diverse shapes, especially thinner. The magnetic microparticles give a weaker flux signal, which is paired with a more sensitive magnetometer. In this way, ReSkin will not affect the objects handled during use. Additionally, ReSkin presents a model that has been generalized and characterized across 18 samples, instead of the typical 1 or assumed 1.
>
>     4. Large area skins: There have been a number of attempts towards developing a large area skin: a number of them based on conductive fabrics with piezoresistive layers[5,6,8] and a few others based on rigid taxels[7] and optical sensors[4]. A number of these solutions however, lack shear sensing capabilities[6,7,8] and conformal contact[7]. Fabric-based solutions often involve a tedious manual sewing process[6] and do not address failure cases and durability. We believe that ReSkin is uniquely positioned to satisfy all of these requirements, and has the potential to be a truly scalable large, soft skin solution.
>
> 2. __Limitations of the sensing skin:__
>     1. Effect of external magnetic fields: The reviewer is correct - external magnetic fields can influence sensor readings. If a strong magnet is in the direct proximity of the magnetometer, the direction of the flux will point towards it. For magnets, the sensor skin will no longer provide the data but the object itself will.
>     2. Multi-point contact: In this current work, the authors only investigated single-touch signals. However, it is possible to define multi-touch if the locations are sufficiently spaced apart (>1cm). If the two contacts are closer than that, they will be registered as one large surface touch. While this is a very interesting question to investigate, it is currently left to future work.
>     3. Latency: The latency upon applying pressure is negligible: the flux readings immediately change with the sensor deformation. Any latency is associated with the material properties, which are more present upon removing pressure. After removing the applied forces, the elastomeric skin takes time to restore to its initial resting state.

---

> > ### Author Response · Authors · 2021-08-20
> > **Response to Reviewer 51C5 (continued)**
> >
> > 3. __Generalization results:__
> >     1. The 18 different sensor skins differ in many ways that are often hard to control, even in a lab environment. The skins were fabricated on different days, at different temperatures and humidities. When mixing the two-part polymer, slight differences in weight can lead to different material properties. Even at the exact same weight, the polymer cross-linking is random and can lead to variation. Adding the magnetic microparticles adds variation in the exact amount that ends up in the molds. The most important factor is magnetic microparticle distribution. The orientation of each particle to itself and others within the skin changes the flux measurement and its response to deformation.
> >
> >     2. For different indenters, previous work has shown that a model trained using data from one indentor can roughly generalize to larger indentors [9]
> >     3. The principles demonstrated in this work are indeed from a very controlled scenario. To test the applicability of our method in a less controlled environment, we performed another experiment where we manually indent the sensor 325 times and run self-supervised adaptation using the triplet loss. We then collect test data using the experimental setup described in Section 4, to evaluate the models. The results are as follows:
> >
> >         *Without adaptation:*
> >
> >         Acc: 79.86%, MSE_xy: 0.676 mm2, MSEFz: 0.268 N2
> >
> >         *With adaptation:*
> >
> >         Acc: 84.87%, MSExy: 0.489 mm2, MSEFz: 0.192 N2
> >
> > 4. __Model-based approaches for calibration:__ The authors have considered model-based approaches to the sensor model. We can model the skin as an elastomer loaded with rigid magnetic particles. However, these models do not account for interactions between magnetic particles. These models also need to be solved with FEA, which furthermore is time-consuming, and assumes a uniform distribution that is not accurate for the skin. Regardless, the authors agree some physics priors, perhaps a simplified physics-model or a handful or pre-solved FEA results, may greatly help constrain the prediction problem and improve results.
> >
> > 5. __Extension to 6D force/torque sensing and shear sensing:__ The skin can sense shear force at this time. To demonstrate this, we performed another experiment with the same experimental setup. Data is collected dynamically by indenting the sensor to a certain depth and dragging it along the sensor. We move in straight lines along x and y directions (as defined by the coordinate system in Section 4) at intervals of 2 mm to cover the entire area of the sensor. This results in a total of over 350K samples. We use an 80-20 split for training and test data, with test data comprising the last 20% of all samples. We use the same network architecture presented in Section 4, predicting (x y Fx Fy Fz) instead of just (x y Fz). For this simple experiment, we get MSE for Fxy: 0.0011 +/- 0.0002 N2, without compromising on prediction of normal force or contact location. Applied shear forces went up to 1 N  in each direction.
> > For torque, we would need to monitor shear forces over time to get a rotation estimate.
> >
> > 6. __Typos and word omissions:__ Thank you for your attention to detail. The authors will carefully review the paper to remove these mistakes.

---

> > > ### Author Response · Authors · 2021-08-20
> > > **Response to Reviewer 51C5 (References)**
> > >
> > >
> > > [1] Donlon, Elliott et al. “GelSlim: A High-Resolution, Compact, Robust, and Calibrated Tactile-sensing Finger.” 2018 IEEE/RSJ International Conference on Intelligent Robots and Systems (IROS) (2018): 1927-1934.
> > >
> > > [2] Wang, Shaoxiong et al. “GelSight Wedge: Measuring High-Resolution 3D Contact Geometry with a Compact Robot Finger.” ArXiv abs/2106.08851 (2021): n. pag.
> > >
> > > [3] Padmanabha, Akhil et al. “OmniTact: A Multi-Directional High-Resolution Touch Sensor.” 2020 IEEE International Conference on Robotics and Automation (ICRA) (2020): 618-624.
> > >
> > > [4] L. Van Duong and V. A. Ho, "Large-Scale Vision-Based Tactile Sensing for Robot Links: Design, Modeling, and Evaluation," in IEEE Transactions on Robotics, vol. 37, no. 2, pp. 390-403, April 2021, doi: 10.1109/TRO.2020.3031251.
> > >
> > > [5] Büscher, Gereon H., et al. "Flexible and stretchable fabric-based tactile sensor." Robotics and Autonomous Systems 63 (2015): 244-252.
> > >
> > > [6] T. Bhattacharjee, A. Jain, S. Vaish, M. D. Killpack and C. C. Kemp, "Tactile sensing over articulated joints with stretchable sensors," 2013 World Haptics Conference (WHC), 2013, pp. 103-108, doi: 10.1109/WHC.2013.6548392.
> > >
> > > [7] Cheng, Gordon, et al. "A comprehensive realization of robot skin: Sensors, sensing, control, and applications." Proceedings of the IEEE 107.10 (2019): 2034-2051.
> > >
> > > [8] Joshua Wade, et al. “A force and thermal sensing skin for robots in human environments”
> > > Robotics and Autonomous Systems, Volume 96, 2017, Pages 1-14, ISSN 0921-8890, doi: https://doi.org/10.1016/j.robot.2017.06.008.
> > >
> > > [9] Hellebrekers, T., Chang, N., Chin, K., Ford, M. J., Kroemer, O., & Majidi, C. (2020). Soft magnetic tactile skin for continuous force and location estimation using neural networks. IEEE Robotics and Automation Letters, 5(3), 3892-3898.
> > >
> > > [10] Lambeta, Mike, et al. "Digit: A novel design for a low-cost compact high-resolution tactile sensor with application to in-hand manipulation." IEEE Robotics and Automation Letters 5.3 (2020): 3838-3845.

---

### Official Review · Reviewer_VwcU · 2021-07-24

**Originality:** Very Good
**Technical Quality:** Excellent
**Clarity Of Presentation:** Excellent
**Impact:** 4

**Recommendation:**

Strong Accept: I recommend accepting the paper and will argue for my recommendation even if other reviewers hold a different opinion.

**Summary:**

The paper proposes a new soft tactile sensor that relies on the change in the magnetic properties of the soft magnetized skin to estimate and localize the applied force. Experiments are carried out to learn the models that map the magnetic field measurements to applied forces, and techniques for generalization when the skin or PCB is replaced are proposed and evaluated. Example applications are described in the paper and demonstrated in the video.

**Issues:**

The video can be shortened. Picking grapes and blueberries is roughly the same, perhaps one of the experiments can be dropped from the video. It would be good to see the sensor readings when the berries are picked up. How noisy are those measurements?

What happens if there is external magnetic field? Is the sensor not useful then anymore? It may be worth adding a paragraph or subsection discussing the limitations of the current design. For example, if one manipulates metal objects that can be magnetized, will this affect the performance? Can you suggest any solutions for this problem?

**Reviewer Expertise:**

Good: General knowledge of the area

**Strengths And Weaknesses:**

Strengths: The proposed sensor has a number of desirable properties: it is small, durable, allows for easy replacement of skin, etc.

Weaknesses: Experiments only seem to evaluate the normal force. Is the estimation of the shear force also reliable? Can one estimate force when the contact surface is large? Can one estimate the skin deformation from the magnetic flux readings?


**Summary Of Recommendation:**

The proposed sensor design may be potentially useful in many applications. The paper showed in detail that the sensor can be utilized at high rate (roughly 400Hz) and with good precision (roughly 1mm). The fact that the sensor design and code are publicly released is a big bonus.

---

> ### Author Response · Authors · 2021-08-20
> **Response to Reviewer VwcU**
>
> We thank the reviewer for positive feedback. We are pleased to confirm that we have done the shear force estimation and discuss all the limitations. We will put these limitations in the final version of the paper.
>
> 1. __Shear force estimation:__ The reviewer is correct that it is possible to measure flux changes when applying shear force. To demonstrate this, we performed another experiment with the same experimental setup. Data is collected dynamically by indenting the sensor to a certain depth and dragging it along the sensor. We move in straight lines along x and y directions (as defined by the coordinate system in Section 4 of the paper) at intervals of 2 mm to cover the entire area of the sensor. This results in a total of over 350K samples. We use an 80-20 split for training and test data, with test data comprising the last 20% of all samples. We use the same network architecture presented in Section 4, predicting (x y Fx Fy Fz) instead of just (x y Fz). For this simple experiment, we get MSE for Fxy: 0.0011 +/- 0.0002 N^2, without compromising on prediction of normal force or contact location. Applied shear forces went up to 1 N  in each direction.
>
> 2. __Multi-point contact and large surface area contact:__ When the contact surface is large there is a corresponding larger shift in magnetic flux. While we believe this is possible, it seems beyond the scope of the current paper.
>
> 3. __Estimating full skin deformation:__ It would be very interesting to fully estimate the skin deformation from the magnetic flux readings. While this would make sensor output more interpretable, at this time, this remains as future work.
>
> 4. __Effect of external magnetic fields and metallic objects:__ The reviewer is correct - external magnetic fields can influence sensor readings. If a strong magnet is in the direct proximity of the magnetometer, the direction of the flux will point towards it. For magnets, the sensor skin will no longer provide the data but the object itself will. But for most practical applications, we have not found small external magnetic fields from motors etc not to be an issue.
>
>     For metal objects that may be ferromagnetic, we found little to no effect. The main reason being that the magnetometer is set at a very high sensitivity that complements a very weak magnetic skin. The magnetic skin will not be able to attract or repel even small metallic objects like screws.
>
>     One way to mitigate these problems is to place another magnetometer out of range from the skin (>5mm). If that magnetometer picks up readings, then the system knows that an external field is now interfering.

---

### Official Review · Reviewer_fsSv · 2021-07-28

**Originality:** Fair
**Technical Quality:** Fair
**Clarity Of Presentation:** Fair
**Impact:** 2

**Recommendation:**

Weak Reject: I recommend rejecting the paper, but will not argue for my recommendation if the majority of other reviewers have a different opinion.

**Summary:**

The work presents a tactile sensor named as ReSkin that can replaces its skin. It is based on magnetic sensing that separates the electronic circuitry from the passive interface, making it easier to replace interfaces as they wear out. A neural network method is also proposed to to learn sensor response models that are variant to across fabrication and time.

**Issues:**

The presentation of the paper can be improved, for example,
1. Experiment Setup is ahead of the sensor models, which makes it confusing to follow.
2. In Section 5, the network architecture can be better illustrated.
3. The contact localization is a easy task for the tactile sensor as it is to detect the contact on the surface, it would be better to include analysis on adaptation for force prediction.

**Reviewer Expertise:**

Very good: Comprehensive knowledge of the area

**Strengths And Weaknesses:**

Strengths:
1. The presented replaceable skin and the model to learn sensor responses across fabrication and time are interesting.
2. Some interesting results are presented in the experiments.

Weaknesses:
The contributions of the paper are not clear as some of the available tactile sensors come with replaceable skins naturally, for example GelSight and GelTip sensors. It is not very convincing to claim that a tactile sensor with replaceable skin is proposed. Also, it is clear how the proposed sensor is different from the magnetic sensors in the literature. It is interesting to learn sensor responses across fabrication and time are interesting but it can be taken separately from the design and fabrication of the ReSkin.
Gomes, D.F., Lin, Z. and Luo, S., 2020, June. GelTip: A finger-shaped optical tactile sensor for robotic manipulation. In 2020 IEEE/RSJ International Conference on Intelligent Robots and Systems (IROS) (pp. 9903-9909). IEEE.


**Summary Of Recommendation:**

Magnetic sensors are one major type of the tactile sensors for robotic applications, it is not clear how the proposed sensor differs from the ones in the literature.

---

> ### Author Response · Authors · 2021-08-20
> **Response to Reviewer fsSv**
>
> Before we answer individual queries, we highlight our response on novelty and contributions in the common response above. In summary, we believe ReSkin is the only viable sensor for robot learning approaches because of its form factor. ReSkin sensors can be put on human gloves which allows collection of object-interaction data in the wild [something which you cannot do with gelsight for example]. And  to the best of the authors’ knowledge, there has been no related work with gelsight-based sensors or any other tactile sensors that characterize the deterioration of the sensor as well as variability across different instances of the sensor.
>
> We now address the rest of the reviewer’s concerns below:
> 1. __Comparison to Gelsight and Geltip:__
> The biggest difference between Gelsight/Geltip and ReSkin is form factor. Gelsights are clunky (>18mm depth) because of the distance required between elastomer and the camera required. So, even if one can replace skin in a gelsight, it is not a viable option for a lot of sensing applications such as force gloves. That said, changing elastomers in a gelsight is a bit more involved process: they require a reflective surface (airbrushed by hand in geltip) or putting a grid to visualize contacts.
> To the best of the authors knowledge, the gelsight-based literature does not state how many sensors have been used to collect the data presented in each paper. Signal quality for gelsight-based sensors is highly dependent on the quality of the elastomer. For example, GelSlim [1] showed that sensor quality begins to deteriorate significantly after 3500 grasps. ReSkin response is much less sensitive to degradation of the elastomer and we show sustained performance over 50k indentations.
>
> 2. __Difference from existing magnetic skins:__ This sensor is different from existing magnetic sensors by using magnet microparticles over mm-scale magnets. By using microparticles, the skin itself can be molded into many shapes and thicknesses. We also modified the training magnets to induce a pattern of edges onto the surface, which we believe reduces variability across fabrication and contributes to the generalizability of the models presented in this paper.
>
> 3. __Generalization of Model Separated from Fabrication:__ We believe that the generalizability of learned models presented in the paper is closely tied to the fabrication techniques used. While some gelsight designs such as DIGIT[2] might have positive  characteristics such as long-lasting, it is just not a viable solution because of the large form factor.
>
> 4. __Self-supervised adaptation for force prediction:__ We would like to highlight we have already done this and the reviewer seems to have missed the experiment. Table 1 in the paper shows an improvement in force prediction as well, when self-supervision is added.
>
>
> [1] Donlon, Elliott et al. “GelSlim: A High-Resolution, Compact, Robust, and Calibrated Tactile-sensing Finger.” 2018 IEEE/RSJ International Conference on Intelligent Robots and Systems (IROS) (2018): 1927-1934.
>
> [2] Lambeta, Mike, et al. "Digit: A novel design for a low-cost compact high-resolution tactile sensor with application to in-hand manipulation." IEEE Robotics and Automation Letters 5.3 (2020): 3838-3845.

---

### Author Response · Authors · 2021-08-20
**Main response**

We thank the reviewers and AC for their feedback. First, we agree with AC that this is not a standard CoRL paper. Therefore, we urge the reviewers and AC to give a second hard look at our paper. Before we start the rebuttal and answer individual queries, we would like to make a passionate appeal (as AC asks for it) for why we believe CoRL is indeed the right venue and why this paper has potential to make a huge impact in the CoRL community.

### Context:

Unlike vision, tactile perception is not the first class citizen in robotics because (a) either the sensors are too expensive (biotac) or (b) the form factor is too clumsy to measure signals in the wild (gelsight and geltip). What currently robot learning and large-scale learning are missing is richness of data which becomes supervision in itself for learning approaches. Learning approaches need an inexpensive, long-lasting and easy-to-use tactile sensor. We believe ReSkin is the only sensor that satisfies all the requirements (See below). We believe our paper has the potential to make a much bigger change in the robot learning community. And we believe the CoRL community needs to know the availability of this sensor.

| | ReSkin (ours) | DIGIT [1] | GelSlim [2] | BioTac [3] | RSkin [4] | Multimodal skin [5] |
|  ----------- | :---------: | :---------: |:---------: |:---------: |:---------: |:---------: |
Type|Magnetic|Optical|Optical | MEMS | Piezoresistive fabric|Sensing chip|
Frequency|~400 Hz|60 Hz|60 Hz|100 Hz|?|250 Hz|
Variable Form Factor|&#9989;|&#10060;|&#10060;|&#10060;|&#9989;|&#10060;|
Low thickness (< 3 mm)|&#9989;|&#10060;|&#10060;|&#10060;|&#9989;|&#9989;|
Low Cost|&#9989;|&#9989;|&#9989;|&#10060;|&#10060;|&#10060;|
Quantified performance over time|&#9989;|&#10060;|&#10060;|&#10060;|&#10060;|&#10060;|
Easy Replaceability|&#9989;|&#9989;|&#9989;|N/A|&#10060;|&#9989;|
Area Coverage|&#9989;|&#10060;|&#10060;|&#10060;|&#9989;|&#9989;|
Durability (>50k contacts)|&#9989;|?|&#10060;|&#9989;|?|?|

### Novelty:

But what about novelty. Reviewers and AC are right (a) Magnetic skins and other sensors have existed. While we do provide some novelty in fabrication, is it novel enough?
(b) self-supervised learning, multi-task learning has existed and there are so many papers. Is there any novelty in learning? So, where does our novelty lie?

First, if this is the yardstick and way we evaluate papers, then award-winning paper [6] should not be novel and should have been rejected. If you look at [6], the network architecture was not new (it was standard ImageNet architecture). So no novelty on the learning side. From a robotics perspective, grasping open-loop experiments was not novel or something to rave about. The novelty of the paper lies in the intersection of Deep Learning and Physical Robots. And it spurned so much followup research. Papers that cross traditional boundaries have a negative bias when judged by individual communities but their novelty lies in the way they bring two fields together.

We believe our novelty is the intersection of self-supervised learning and soft sensing. It might seem that there are already papers at the intersection of tactile and learning but we want to highlight that all those papers are at the application interface -- the sensors are already working and to make the application work (marble juggling etc.) learning is used. On the other hand, in our case, the sensor signal by itself is not that meaningful and useful. The learned models fail to reproduce across multiple fabrications and over wear and tear. It is the large-scale and self-supervised learning that actually makes the sensor usable.

Therefore, our core contribution is to advocate the use of inexpensive materials and low-form factor sensing but overcome wear-tear and fabrication issues via use of large-scale and self-supervised learning. More specifically,

1. Our paper introduces novel fabrication techniques to reduce the thickness, introduce variable form factors [by using mm particles] and reduce noise signal.
2. Our paper is the first to perform large-scale empirical analysis to understand the relationship between wear/tear and learned models.
3. Our paper is first to demonstrate that fabrication issues and inconsistencies can be dealt with via a simple self-supervised scheme and a few hundred samples. Our SSL calibration technique could be useful for other sensors/tasks as well.

But above all, we believe our sensor is the only sensor which satisfies all the requirements to emerge as the only viable sensor for learning applications.

...

---

> ### Author Response · Authors · 2021-08-20
> **Main Response (continued)**
>
> ### Are Self-supervised results statistically significant?
>
> We performed paired t-tests using data for "Multi-sensor without triplet loss" and "Multi-sensor with triplet loss, adapted using 390 indentations", to evaluate the statistical significance of our results. We find t=4.575 and t=4.596 for MSExy and MSEFz respectively, which are both significantly higher than t.9995=3.965 for 𝜈 = 17. This clearly highlights that our results are statistically significant.
>
> ### Other significant additions (details below):
>
> We have also added two new experiments which highlight that ReSkin can be used for shear force estimation and that self-supervised calibration works even in the wild where the skin is manually pressed to collect the data. We hope reviewers 51C5 and fsSv can look at these results along with other discussions to re-evaluate the paper and its contributions.
>
> [1] Lambeta, Mike, et al. "Digit: A novel design for a low-cost compact high-resolution tactile sensor with application to in-hand manipulation." IEEE Robotics and Automation Letters 5.3 (2020): 3838-3845.
>
> [2] Donlon, Elliott et al. “GelSlim: A High-Resolution, Compact, Robust, and Calibrated Tactile-sensing Finger.” 2018 IEEE/RSJ International Conference on Intelligent Robots and Systems (IROS) (2018): 1927-1934.
>
> [3] Sundaralingam, Balakumar, et al. "Robust learning of tactile force estimation through robot interaction." 2019 International Conference on Robotics and Automation (ICRA). IEEE, 2019.
>
> [4] T. Bhattacharjee, A. Jain, S. Vaish, M. D. Killpack and C. C. Kemp, "Tactile sensing over articulated joints with stretchable sensors," 2013 World Haptics Conference (WHC), 2013, pp. 103-108, doi: 10.1109/WHC.2013.6548392.
>
> [5] Cheng, Gordon, et al. "A comprehensive realization of robot skin: Sensors, sensing, control, and applications." Proceedings of the IEEE 107.10 (2019): 2034-2051.
>
> [6] L. Pinto and A. Gupta, "Supersizing self-supervision: Learning to grasp from 50K tries and 700 robot hours," 2016 IEEE International Conference on Robotics and Automation (ICRA), 2016, pp. 3406-3413, doi: 10.1109/ICRA.2016.7487517.

---

### Author Response · Authors · 2021-08-30
**Request for discussion**

Dear AC and Reviewers,

We were hoping for a discussion:

1. With Reviewer 51C5 in light of the new experiments and comparisons to state-of-the-art
2. With AC with respect to our arguments for applicability to CoRL and novelty.
3. With Reviewer fsSv in light of the comparison to others and reread of force adaptation result.

While we haven’t had any discussions yet, we are still available to answer more questions today. We sincerely believe this paper has potential to have a big impact in the robot learning community. We will upload an updated version of the paper later today.

Thanks

Authors

---

### Meta-Review · Area_Chair_ikBu · 2021-08-17

**Recommendation:** Accept (Oral)
**Confidence:** 4

**Metareview:**

This paper proposes a magnetic, replaceable soft skin sensor with novel form factor. The paper explains the design for improving and producing more skin sensors for use in robotic manipulation with a specific focus towards enabling the use of learning-based methods. The paper uses learning to create sensor processing that is robust across the variations of different manufactured skin elements.

Furthermore the authors argue for the need of tactile sensing as a common modality in robot manipulation to improve performance when working outside of structured environments. Furthermore, they explain why learning is well suited for use with tactile sensors.

I agree with this importance and the recent growth in tactile sensing for learning-based manipulation provides further evidence for its importance. As such, I think this paper would be an interesting inclusion at CoRL and spark helpful discussions.

Finally, I agree with reviewer 51C5 and caution the authors against overly broad claims such as "But above all, we believe our sensor is the only sensor which satisfies all the requirements to emerge as the only viable sensor for learning applications." Indeed the citations being given show other sensors are viable for learning. Perhaps the authors mean that the durability and replacement of the sensors make them more suitable to long term use and abuse that is common in reinforcement learning techniques. However, between safe RL methods and other more careful data-collection used in previously published learning-based work for tactile sensing, other sensors are clearly appropriate for robot learning.

I believe the paper would be better received and appreciated if the authors offer this as an additional sensor that can often be useful on its own, but can also complement other existing sensors (e.g. I could imagine using BioTacs or GelSight sensors on the fingertips of a robot with the ReSkin covering the rest of the robot hand and arm.

---

### Decision · Program_Chairs · 2021-09-13

**Decision:**

Accept (Oral)

**Comment:**

This paper proposes a magnetic, replaceable soft skin sensor with novel form factor. The paper explains the design for improving and producing more skin sensors for use in robotic manipulation with a specific focus towards enabling the use of learning-based methods. The paper uses learning to create sensor processing that is robust across the variations of different manufactured skin elements.

Furthermore the authors argue for the need of tactile sensing as a common modality in robot manipulation to improve performance when working outside of structured environments. Furthermore, they explain why learning is well suited for use with tactile sensors.

I agree with this importance and the recent growth in tactile sensing for learning-based manipulation provides further evidence for its importance. As such, I think this paper would be an interesting inclusion at CoRL and spark helpful discussions.

Finally, I agree with reviewer 51C5 and caution the authors against overly broad claims such as "But above all, we believe our sensor is the only sensor which satisfies all the requirements to emerge as the only viable sensor for learning applications." Indeed the citations being given show other sensors are viable for learning. Perhaps the authors mean that the durability and replacement of the sensors make them more suitable to long term use and abuse that is common in reinforcement learning techniques. However, between safe RL methods and other more careful data-collection used in previously published learning-based work for tactile sensing, other sensors are clearly appropriate for robot learning.

I believe the paper would be better received and appreciated if the authors offer this as an additional sensor that can often be useful on its own, but can also complement other existing sensors (e.g. I could imagine using BioTacs or GelSight sensors on the fingertips of a robot with the ReSkin covering the rest of the robot hand and arm.